# First Experiences with Over-the-Counter Hearing Aids for Mild-to-Moderate Hearing Loss: A Case Report

**DOI:** 10.3390/pharmacy12010004

**Published:** 2023-12-28

**Authors:** Lucas A. Berenbrok, Annie Duchen, Tony Cyr, Elaine Mormer

**Affiliations:** 1Department of Pharmacy and Therapeutics, School of Pharmacy, University of Pittsburgh, Pittsburgh, PA 15261, USA; 2WS Audiology, Rockville, MD 20850, USA; anne.duchen@wsa.com (A.D.); anthony.cyr@wsa.com (T.C.); 3Department of Communication Science and Disorders, School of Health and Rehabilitation Sciences, University of Pittsburgh, Pittsburgh, PA 15260, USA; emormer@pitt.edu

**Keywords:** over-the-counter hearing aids, hearing healthcare, community pharmacy, public health, hearing loss, self-care

## Abstract

This patient case report describes a first experience in late 2022 and early 2023 with over-the-counter (OTC) hearing aids for a 71-year-old male with self-perceived, age-related hearing loss. The patient reported no “red flag” medical conditions that would preclude him from safely using an OTC hearing aid device. After also meeting inclusionary criteria required to be printed on the device label, the patient was offered FDA registered OTC hearing aids. The first device pair was returned due to malfunction. The second device pair was an in-the-canal style, black in color, and powered by disposable batteries. He required help setting up the device from his spouse, an audiologist, and a pharmacist. Improved scores on the Self-Assessment of Communication and Significant Other Assessment of Communication were noted from the patient and his spouse. The patient continued to use the second device pair for 6 months after first use with no additional help. Our experience supports the pharmacist’s role in identifying appropriate candidates for OTC hearing aids, helping patients select a device, and supporting device setup and self-fitting processes at community pharmacies. Further experiences are needed to demonstrate how pharmacists can support OTC hearing aid purchases at community pharmacies.

## 1. Introduction

The National Institutes of Health estimates that nearly 28.8 million adults in the U.S. could benefit from using hearing aids [1]. However, fewer than 30% of adults aged 70 and older with hearing loss have ever used hearing aids, and only 16% of adults aged 20–69 with hearing loss have used them [1]. Such low rates of use suggest that there are barriers to adoption. Accessibility and affordability were identified as major barriers by the President’s Council of Advisors on Science and Technology in 2015 [2]. In response, over-the-counter (OTC) hearing aids were proposed as a solution to provide consumers with greater access to hearing aids at lesser costs. In August 2022, the U.S. Food and Drug Administration (FDA) formalized this solution by releasing final regulations that established a new category of hearing aids to be sold OTC [3]. However, in the year after these final regulations went into effect, the American Speech-Language-Hearing Association reported that only 2% of persons surveyed with hearing loss have purchased an OTC hearing aid and only 4% stated an intention to purchase one [4]. Such low adoption rates of OTC hearing aids suggest that additional measures are needed to increase hearing aid use.

Historically, hearing aids have been available to persons with hearing loss only after medical evaluation by a physician and audiological examination by a licensed hearing care professional. Hearing care professionals include audiologists and hearing aid dispensers [5]. Although medical clearance from a physician was historically required by law for use of traditional, now known as prescription hearing aids, the FDA has not recently enforced this requirement. This has allowed patients direct access to licensed hearing care professionals [6]. Now that OTC hearing aids can be sold without medical clearance and without examination by hearing care professionals, consumers can make independent decisions on their candidacy for purchasing OTC hearing aid devices at retail locations. When purchased at community pharmacies, patient decisions for hearing self-care can be guided by pharmacists. As experts in self-care, pharmacists can assess patients for exclusions to using OTC hearing aids, assist patients in selecting a device, and refer patients to hearing care professionals when further management is needed [7].

After purchase, consumers may independently set up their OTC hearing aids. For some models, setup will be aided by a mobile application for smart phones. Apps like these help patients complete a self-fitting process to self-test their hearing or to adjust the device to their individual needs, without a licensed hearing care professional. Interestingly, a recent study by DeSousa et al. showed that OTC hearing aids were equally effective when fit by patients vs. audiologists [8]. However, the process of self-managing OTC hearing aids requires that consumers have basic health literacy and digital literacy to be successful. Health literacy is a person’s ability to “gain access to, understand and use information in ways which promote and maintain good health [9]”. Digital literacy is defined as “the ability to use information and communication technologies to find, evaluate, create, and communicate information, requiring both cognitive and technical skills [10]”. Expected self-management functions for successful use of OTC hearing aids include self-determining candidacy for safe use; product selection based on device features and functions; and device set up, fitting, and maintenance. Patient initial acceptance and continued use of an OTC hearing aid device could be enhanced or diminished based upon their experience with any of these self-managed functions. Therefore, pharmacists may play a role in supporting patients purchasing OTC hearing aids given their proximity to device sales at community pharmacies [11]. To further guide pharmacists offering or preparing to offer OTC hearing aids at the community pharmacy, a list of 26 pharmacist competency statements for OTC hearing aids is available [7].

The objective of this case report is to describe first experiences with OTC hearing aids for a male with age-related hearing loss. The patient and his spouse agreed to participate and approved his inclusion in this report.

## 2. Materials and Methods

### Clinical Assessment for OTC Hearing Aid Use

Patient 1, a 71-year-old retired male, was approached by pharmacist author (L.A.B.) in November 2022 because of his known intent to purchase OTC hearing aids for self-perceived, age-related hearing loss. The patient first noted his hearing loss 20 years ago. At the time of this case report, he noted difficulty understanding speech in conversation, especially when background noise was present. His spouse also noted that he watched television too loudly for others with normal hearing. Both symptoms suggested perceived mild to moderate hearing loss. A full list of symptoms is required to be present on the outside of the box labeling of OTC hearing aids per FDA regulations [12]. The patient’s primary goals were to obtain OTC hearing aids quickly given his excitement for an affordable hearing solution. He initially sought a style that would fit in the ear canal versus behind the ear. He also sought a rechargeable battery feature.

Prior to presenting an OTC hearing aid device to the patient, the audiologist author (E.M.) performed a hearing test to confirm the patient’s degree of hearing loss. The hearing test was completed on the University of Pittsburgh campus in a sound treated, audiometric booth. The audiologist checked the patient’s ear canals with an otoscope and noted no wax build-up and no occlusion. To his best recollection, the patient’s last hearing test was 45 years ago. The patient owned an Apple iPhone which was relevant to the device setup and self-fitting processes.

The patient was confirmed to have hearing loss at frequencies typical of age-related hearing loss, consistent with a mild-to-moderate degree at higher frequencies. In addition, the pharmacist author asked the patient about “red flag” medical conditions which can be found in the labeling of OTC hearing aid device as mandated by FDA regulations (Figure 1). “Red flag” conditions are like exclusions to self-care for nonprescription drugs. The patient denied all “red flag” conditions that would preclude him from using OTC hearing aids without medical supervision. Ultimately, the patient was determined to be an appropriate candidate for OTC hearing aid use.

Following confirmation of the patient’s eligibility for OTC hearing aid use, the pharmacist author asked a series of questions to quantify perceived disability connected with his hearing loss using the Self-Assessment of Communication (SAC) tool [13]. His spouse independently completed an accompanying measurement tool, the Significant Other Assessment of Communication (SOAC). The SAC is a 10-item screening measure probing an individual’s self-report of communication difficulty, handicap, and perceptions of others across various listening situations. The SOAC is a questionnaire that probes the perceptions of those who are significant others of the individual experiencing hearing loss, using the same 10 items in the SAC. The SAC and SOAC tools were used with permission from the creator.

## 3. Results

### Patient Experience with OTC Hearing Aid Devices

The first device pair was purchased online by authors (E.M. and L.A.B.) and presented to the patient. The devices were completely in-the-canal, FDA registered OTC hearing aids. They were silver in color and rechargeable. The patient was observed unboxing the devices and setting up the mobile application to perform the self-fitting process immediately after his audiological examination. The setup of the devices took nearly one hour. The patient was assisted in setting up the devices by his spouse and by authors (E.M. and L.A.B.). The professional skill set of the audiologist was not offered to the patient during device set up. The patient wore these hearing aids for the next week. After seven days, the patient called the pharmacist author (L.A.B.) to report that the right hearing aid would no longer hold a charge. Thereafter, L.A.B. met with the patient in-person to further troubleshoot. The device was deemed nonfunctional and was ultimately returned directly to the manufacturer for a full refund.

A second set of FDA registered OTC devices (Sony CRE-C10 Self-Fitting OTC hearing aids [Sony]) was offered to the patient approximately one month after the first. This second set of devices was completely in-the-canal, FDA-cleared, black in color, and used disposable batteries. The device required access to a smartphone to download the manufacturer’s proprietary mobile application, necessary to complete the self-fitting process. Again, the patient required technical assistance from authors (E.M. and L.A.B) during device setup to download and navigate the mobile app, although the audiologist primarily observed. In February 2023, following two months of continuous wear, the patient reported satisfaction with the device. For most items of the SAC and SOAC, improvement was noted. Improved total scores on the SAC and SOAC, aided by the device, are presented in Table 1. These improved scores reflect a lower percentage of time in which the patient’s hearing loss was perceived to cause problems. The patient continued to use the device for 6 months after initial setup and self-fitting, with no additional help.

## 4. Discussion

OTC hearing aids are expected to make hearing devices more accessible and more affordable for millions of Americans with hearing loss. But how persons with hearing loss will seek and adopt OTC hearing aids remains largely unknown. Here we present our first experiences with OTC hearing aid devices to show pharmacists, audiologists, and hearing device manufacturers how OTC devices may be used by consumers.

OTC hearing aids present an opportunity for persons with hearing loss to seek care sooner. On average, it takes four years for a person to visit a hearing care professional or a physician after first becoming aware of their hearing loss [14]. Although the reason for delay in seeking hearing aids is multifactorial, direct OTC access to hearing aids can simplify traditional pathways and in turn, accelerate a person’s journey to hearing care. Patient 1 was agreeable to participating in this case report because he perceived that OTC hearing aids would be easier to access than prescription hearing aids. To illustrate this point further, there were approximately 2906 hearing aid dispensing offices in the U.S. in 2022, compared to 61,715 community pharmacy locations in October 2020 [15,16]. Therefore, adding community pharmacies as locations in which consumers can seek and purchase hearing aids will expand public access to hearing care.

Our experience with Patient 1 suggests that pharmacists can accurately identify appropriate candidates for OTC hearing aids. In our report, the pharmacist author correctly identified an individual with perceived mild-to-moderate hearing loss. Although we confirmed the patient’s severity of hearing loss via a hearing test performed by the licensed audiologist, this step was performed solely for the purposes of proving the concept. Federal law and regulation do not require a hearing examination prior to purchase or set-up of OTC hearing aids. In addition, the pharmacist author successfully collected information about the patient’s perception of hearing problems using the SAC tool. Pharmacists may consider using this tool to collect and track patient outcomes after using OTC hearing aids. When outcomes remain unchanged, pharmacists should consider referring patients to local hearing care professionals for additional evaluation and comprehensive care [17]. In addition to assessing effectiveness, pharmacists should also be familiar with product labeling to assess safety prior to sale. In our report, the pharmacist correctly used product labeling to ensure that the patient did not present with “red flag” conditions, which would exclude him from hearing self-care.

Patient 1’s experience also suggests that patients purchasing OTC hearing aid devices will require differing levels of support with the initial self-fitting process and maintenance that follows. For both devices, Patient 1 required support to download and navigate the mobile applications required to set up the devices. Although an estimated 79% of adults 50 and older use a smartphone, only 51% of older adults who own a smartphone reported downloading or purchasing an app in the past three months [18]. Downloading an app is an essential function of most self-fitting OTC hearing aid models. As a result, pharmacists and supporting staff should consider making themselves available to support the self-fitting process for OTC hearing aids purchased at community pharmacies. Interestingly, these roles were anticipated in 2020 by a stakeholder panel which included pharmacists, audiologists, device manufacturers, and persons with hearing loss who brainstormed and agreed upon pharmacist competencies for helping people seeking OTC hearing aids at the community pharmacy [7]. Relevant pharmacist competency statements include “Access manufacturer resources to assist patients with the care and operation of hearing devices sold at the pharmacy [7]”.

In addition to older adults, who have the highest prevalence of hearing loss [19], technical support may also be needed for younger individuals with low-health literacy or digital literacy. However, these individuals may also seek technical support from family, friends, hearing care providers, or directly from manufacturers. Although many device manufacturers offer remote support after product purchase, community pharmacists can uniquely offer in-person support to troubleshoot devices, as they do with blood pressure monitors and glucometers. But it remains unknown if in-person support from pharmacists positively enhances the patient’s experience with OTC hearing aid devices.

Pharmacists should also be prepared to help patients select OTC hearing aids stocked at the community pharmacy based on patient preferences for device features and functions [7]. In the case of Patient 1, he initially sought in-the-canal, rechargeable devices to conceal his hearing loss and to save money, respectively. However, he was ultimately satisfied with a device using disposable batteries, which offered more flexibility during outdoor recreation. Lastly, this case offers insight into returning devices due to malfunction or failure to meet patient expectations. Pharmacies should be prepared to handle consumer requests to return devices for the former, and pharmacists should consider referral to audiologists for the latter. Notably, federal regulations do not require manufacturers or sellers to accept returns [3].

## 5. Conclusions

This patient case report describes a first experience with OTC hearing aids for a male with age-related hearing loss. Our experience supports the pharmacist’s role in identifying appropriate candidates for OTC hearing aids, helping patients select a device at community pharmacies, and supporting device setup and self-fitting processes. Further experiences are needed to demonstrate how pharmacists can support OTC hearing aid purchases at community pharmacies.

## Figures and Tables

**Figure 1 pharmacy-12-00004-f001:**
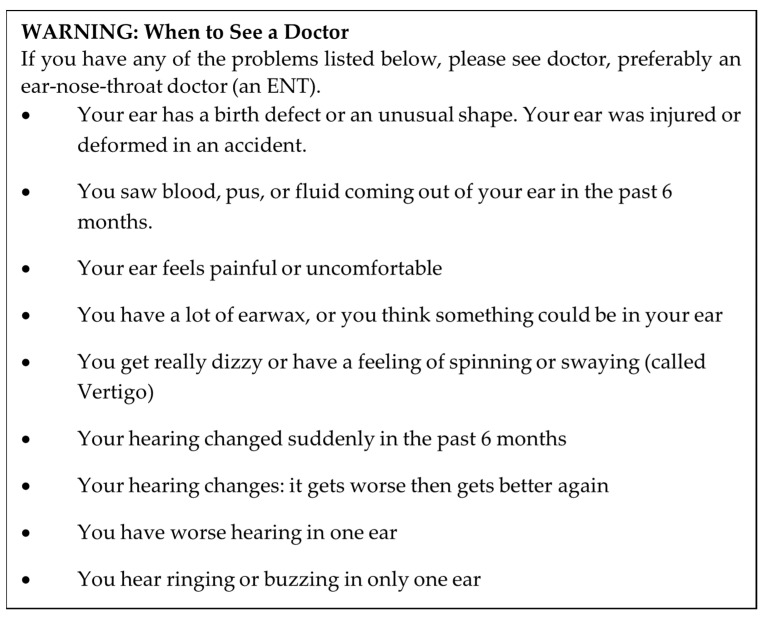
“Red flag” medical conditions from the FDA’s final rule establishing OTC hearing aids [3].

**Table 1 pharmacy-12-00004-t001:** Pre- and post-assessment of hearing loss by patient and significant other.

Self-Assessment of Communication Items and Significant Other Assessment of Communication [13]	Patient		Patient’s Spouse	
	Before OTC hearing aid fitting *	After 2 months of OTC hearing aid use **	Before OTC hearing aid fitting *	After 2 months of OTC hearing aid use **
1. Do you experience communication difficulties in situations when speaking with one other person? (For example, at home, at work, in a social situation, with a waitress, a store clerk, with a spouse, boss, etc.)	Almost never (1)	Almost never (1)	About half of the time (3)	Almost never (1)
2. Do you experience communication difficulties in situations when conversing with a small group of several persons? (For example, with friends or family, co-workers, in meetings or casual conversations, over dinner or while playing cards, etc.)	Occasionally (2)	Almost never (1)	About half of the time (3)	Occasionally (2)
3. Do you experience communication difficulties while listening to someone speak to a large group? (For example, at a church or in a civic meeting, in a fraternal or women’s club, at an educational lecture)	Almost never (1)	Almost never (1)	Frequently (4)	Occasionally (2)
4. Do you experience communication difficulties while participating in various types of entertainment? (For example, movies, TV, radio, plays, night clubs, musical entertainment, etc.)	Frequently (4)	Occasionally (2)	Practically always (5)	Occasionally (2)
5. Do you experience communication difficulties when you are in an unfavorable listening environment? (For example, at a noisy party, where there is background music, when riding in an auto or bus, when someone whispers or talks from across the room, etc.)	Practically always (5)	Occasionally (2)	Practically always (5)	Almost never (1)
6. Do you experience communication difficulties when using or listening to various communication devices? (For example, telephone, telephone ring, doorbell, public address system, warning signals, alarms, etc.)	Almost never (1)	Almost never (1)	About half of the time (3)	Almost never (1)
7. Do you feel that any difficulty with your hearing limits or hampers your personal or social life?	Occasionally (2)	Almost never (1)	About half of the time (3)	Almost never (1)
8. Does any problem or difficulty with your hearing upset you?	Occasionally (2)	Almost never (1)	About half of the time (3)	Almost never (1)
9. Do others suggest that you have a hearing problem?	Almost never (1)	Almost never (1)	Frequently (4)	Almost never (1)
10. Do others leave you out of conversations or become annoyed because of your hearing?	Almost never (1)	Almost never (1)	Almost never (1)	Almost never (1)
Total	25%	5%	60%	7.5%

Scale: Almost never (1); Occasionally (2); About half the time (3); Frequently (4); Practically always (5). * Answered without aid, November 2022. ** Answered aided with Sony CRE-C10 Self-Fitting OTC hearing aids, February 2023.

## Data Availability

The data presented in this study are available on request from the corresponding author.

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
