# Peer review of "First Experiences with Over-the-Counter Hearing Aids for Mild-to-Moderate Hearing Loss: A Case Report"

_pharmacy, 2023, doi:10.3390/pharmacy12010004_

Round 1

Reviewer 1 Report (Previous Reviewer 2)

Comments and Suggestions for Authors

My comments and concerns have been addressed.

Author Response

Thank you for your timely response to our revisions.

Reviewer 2 Report (New Reviewer)

Comments and Suggestions for Authors

This case report describes a hearing impaired aged individual who wanted to reach a fast hearing rehabilitation by hearing aids (HA) over-the-counter (OTC) guided by a pharmacist. The 71-year-old male receives an in-the-canal device with disposable batteries. Though one of the HAs appears defect the patients get a new set of HAs after 1 month. The patient and his spouse are subjected to questionnaires at the start of wearing the HAs and 2 months later. From the answers of these questionnaires it is obvious that the HAs give the patient a better hearing situation. The answers of the spouse also support the traditional observation that that the people in the surrounding of the hearing impaired note that the hearing impaired has more pronounced listening difficulties than the patients grade themselves. 

My opinion is that this case report strengthens the fact that hearing aids can be supplied over-the-counter and considerably improve the hearing. Then aged people my need some help of spouses, relatives etc to handle the digital issue concerning down-loading the app or other adjusting maneouvres.

I find this case report interesting and I strongly recommend it for publication in Pharmacy. I am however somewhat amazed that still there are no case-controlled studies performed on this subject.

Author Response

Thank you for your timely review of our manuscript.

Reviewer 3 Report (New Reviewer)

Comments and Suggestions for Authors

The study is extremely interesting. But when the patient has to set up the first devices, in lines 124-128 we read "The patient was assisted in setting up the devices by the spouse and the authors", that is, a pharmacist and an audiologist. In the setting up of the second devices, the patient was helped only in the installation part of the management software.

The article highlights the crucial and extremely useful role that pharmacists can have in supporting patients in the choice and initial care of OTC devices. But, in this case, it seems that at least initially the presence of an audiologist was necessary. And this contradicts the conclusion that only the pharmacist can alone manage the OTC devices. Maybe more experience is needed for the set up? A different formation? It is not clear from the article whether the pharmacist alone would have been able to help with the selection, assistance with the software, configuration, and set up  of the OTC devices.

The type of support provided by the audiologist must be clarified. And if the audiologist had an important role, the conclusion must be changed: in this case the pharmacist needed the intervention of an audiologist.

Author Response

The study is extremely interesting. But when the patient has to set up the first devices, in lines 124-128 we read "The patient was assisted in setting up the devices by the spouse and the authors", that is, a pharmacist and an audiologist. In the setting up of the second devices, the patient was helped only in the installation part of the management software.

Thank you for pointing out this similarity. We updated lines 138-139 to emphasize that the patient needed help setting up the second device, just like the first.

The article highlights the crucial and extremely useful role that pharmacists can have in supporting patients in the choice and initial care of OTC devices. But, in this case, it seems that at least initially the presence of an audiologist was necessary. And this contradicts the conclusion that only the pharmacist can alone manage the OTC devices. Maybe more experience is needed for the set up? A different formation? It is not clear from the article whether the pharmacist alone would have been able to help with the selection, assistance with the software, configuration, and set up  of the OTC devices.

Thank you for this comment. We updated lines 128-129 to highlight that the professional skill set of the audiologist was not offered to the patient during setup of the first device. Furthermore, the audiologist primarily observed the patient setting up the second device. The help needed from the patient was related to technology and digital literacy, not audiological in nature. Lines 139-141 have been updated accordingly.

The type of support provided by the audiologist must be clarified. And if the audiologist had an important role, the conclusion must be changed: in this case the pharmacist needed the intervention of an audiologist.

Thank you for pointing this out. The audiologist did not have an important role in device setup.

Round 2

Reviewer 3 Report (New Reviewer)

Comments and Suggestions for Authors

I thank the authors for the clarifications provided.

The article in this form is suitable for publication.

This manuscript is a resubmission of an earlier submission. The following is a list of the peer review reports and author responses from that submission.

Round 1

Reviewer 1 Report

Comments and Suggestions for Authors

A) The case is the "optimal patient with the optimal spouse and the optimal hearing loss". Two months is a too short period to evaluate benefits and there are no sufficient considerations over the two points which moved the patient and his spouse to seek for hearing aid after 20 yrs : 1) difficulty understanding speech in 77 conversation, especially when background noise was present 2) watched television too loudly. Questions 2-5 in chosen questionnaires are still the worst after OTC HA. Might be necessary add COSI (client oriented scale of improvement) to better assess targets and results of chosen  hearing aids ??

B) Pharmacist's role is not so clear : better rielaborate with an adequate number of cases (no less than 10, better more) to give more space to Pharmacist connection with the audiologist. The treatment is still up to audiologist's exam so different types of thresholds might help pharmacist to address patients in coping with audiologists suggestions. 

C) should be added "failed" OTC hearing aid setting cases : is not only a matter of app use competence, rather of patient's compliance to hearing through a device and relative impact on QoL , for instance while driving cars, listening  in close vs open spaces , phone calling ... So better clear when pharmacist should stop the patient and advice to seek again for audiologist's check to modify chosen OTC hearing aids 

Reviewer 2 Report

Comments and Suggestions for Authors

This manuscript is based on the authors' experience assisting a single individual using an over-the-counter (OTC) hearing aid.  The manuscript provides a review of information about OTC hearing aids, which were approved by the FDA only recently. That review would likely be useful for pharmacists who are are not already familiar with hearing difficulties, disorders, and devices.  That is a strength.  The manuscript also includes the responses of one participant and his spouse to 10-item questionnaires about hearing difficulties, administered pre- and post-fitting.  These results are not novel or unexpected, but providing data of this type is not the main point of the manuscript.  Instead, the manuscript seems intended to describe what the authors consider to be a research project.  Evidence for that can be found in the language they use:  the authors consistently refer to themselves as "researchers", and they use phrases like "Further research is needed".  In the Discussion and Conclusions, the emphasis is not on the participant's hearing, but on the role of pharmacists.  If what was done was research rather than clinical practice, I disagree with the statement that IRB approval was not required.  Unless the IRB itself made that determination (this was not indicated), proceeding without IRB approval was a major error, in my opinion.